# Scientific and Practical Challenges for the Development of a New Approach to the Simulation of Remanufacturing

Pawel Pawlewski 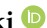

Faculty of Engineering Management, Poznan University of Technology, 60-965 Poznań, Poland; pawel.pawlewski@put.poznan.pl

**Abstract:** This article explores the scientific and practical challenges associated with developing simulation modeling methods for remanufacturing within a circular economy framework. It aims to define this concept and identify the key hurdles that need to be overcome for its successful implementation. According to the six principles of sustainable development, the key pillar is remanufacturing. Remanufacturing helps the environment in several different ways, including: saving energy, the conservation of raw materials, the conservation of space, landfills, the reduction of air pollution, and greater fuel efficiency. This process closes the loop in the supply chain, exemplifying the principles of a circular economy. The research methods used are primarily the analysis and criticism of literature, document examination—especially in relation to existing simulation programs and analysis—the logical construction method, and the heuristic method, used to define concept of simulation modelling. In response to scientific and practical challenges, the concept of a new modeling method was defined and presented. This concept uses the legacy of Lean and the author's original ideas regarding the structuring of the remanufacturing factory and processes. The main contribution of this study is integration, embedding this concept into the simulation software. A comparison with existing solutions and the advantages of the new concept are also included in the article.

**Keywords:** environment sustainability; remanufacturing; digital twin; simulation

## 1. Introduction

In the European Union alone, more than 2.5 billion pieces of waste are produced annually [1]. This is the effect of the linear economy, which in turn results in the total consumption of resources whose availability is limited. This economy relies on large quantities of cheap and readily available materials and energy. It is proposed to change this model, striving for a circular economy. This model prioritizes reusing, repairing, refurbishing, and recycling existing materials and products for as long as possible, minimizing waste and extending the lifespan of resources [2–4]. A useful framework to help reduce environmental impact and improve sustainability was defined. The "6Rs" of sustainability are Reduce, Reuse, Recycle, Refuse, Rethink, and Repair [3]. At the same time, we are in the middle of the fourth industrial revolution, which consists of the broadly understood robotization of material industrial production and which requires a significant change in the organization of work and the appropriate digitization of supply chains. Industry 4.0 is an umbrella term, encompassing a range of digital technologies that work together seamlessly. These include the Internet of Things (IoT), Cyber-Physical Systems (CPS), Big Data, Data Analytics, Digital Twins, Digital Shadows, and Human–Robot Collaboration (HRC) [5]. Industry 4.0, also known as the fourth industrial revolution, has ushered in a new era of manufacturing with smart factories. These intelligent facilities leverage interconnected machines, data analytics, and advanced automation to achieve superior flexibility and resource optimization, ultimately enabling enhanced customer service. The key principles for Industry 4.0 are as follows [6]:

- Transforming into digital, flexible factories unlocks continuous, real-time communication across workstations and tools. This seamless integration streamlines production lines and supply chains, enabling greater agility and responsiveness.
- Simulation and data processing tools empower employees to gain a deeper understanding of industrial conditions and processes. This is achieved by collecting and analyzing assembly line data, which is then used for modeling and testing. This allows employees to visualize potential improvements and optimize production.
- Communication networks become the backbone of energy and resource efficiency in factories. These networks enable continuous, real-time information exchange, allowing for the perfect coordination of needs and availability.

Supply chains and factories face inherent complexity due to both product complexity and production technology. The way processes are organized in a factory adds another layer to this complexity. These systems are also highly dynamic and are constantly adapting to external forces, such as market changes that require product adjustments. Internally, they evolve due to factors such as new product introductions, product recalls and modifications in the organization of material flow. When studying complex systems characterized by intricate details and dynamic cause-and-effect relationships, simulation stands out as a powerful tool. Its ability to manipulate space and time allows us to untangle these relationships, even when they are distant in space and involve intricate feedback loops [7].

In an Industry 4.0 smart factory, every device—from simple controllers to industrial robots—has its own software model called a "digital twin". The twin is constantly fed with data from its physical counterpart, so it "knows" what state it is in. However, a digital twin has much more data—the entire history of the device's behavior, including access to other twins and their data. As a result, it can be used to simulate the different situations in which this device may find itself and to predict the effects of various events. By analyzing digital twins, you can, for example, investigate what device number 237 will do when device number 182 breaks down and fails. Device 237 can be an exit gate, transformer or crane, while device 182 can be a gas valve, an autonomous conveyor, or a furnace. This situation is shown in Figure 1—"Digital Twin (1)" at level 0—representing resources. At the factory level (level 2)—"Digital Twin (2)"—a different approach is required, which is process-based. This also applies to remanufacturing, recycling, etc. At this level, the support of existing programs on the market in the field of simulation modeling methodologies using Lean is insignificant, which is the subject of Section 4 of this article.

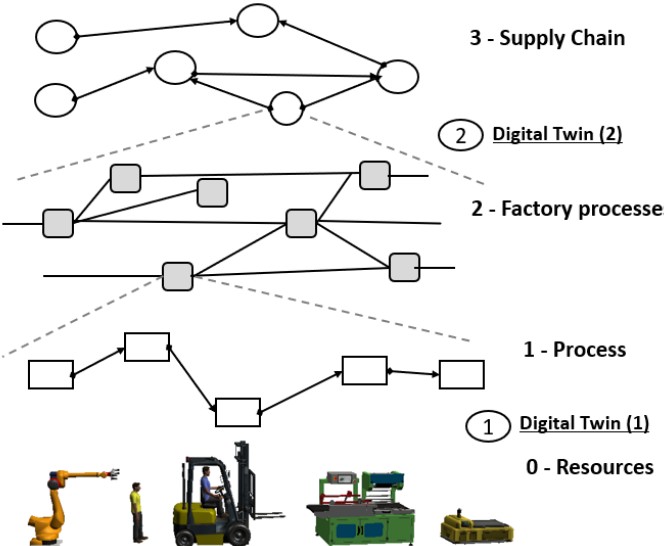

**Figure 1.** Context of the simulation digital twin concept.

This article delves into the new concept of simulation modeling for remanufacturing within a circular economy framework. It aims to define this approach and identify the key scientific and practical challenges that need to be addressed for its successful implementation. The highlights of this article are:

- The definition of scientific challenges in the context of solutions offered by existing approaches to simulation modeling, specifically the postulate of 1:1 modeling enabling Lean analyses;
- By identifying the practical challenges that can shorten modeling time, we can empower production engineers, lean specialists, and logistics specialists to dedicate more time to core production activities.

The contributions from the author is the exploration of:

- The concept of using Lean methods in simulation modeling;
- The concept of structuring a remanufacturing company;
- The concept of structuring remanufacturing processes.

The research methods used are primarily:

- The method of analysis and criticism of literature (which is the subject of Sections 2 and 3);
- The document examination method (especially with regard to existing simulation software);
- The analysis and logical construction method and heuristic method, which are used in Section 4.

The logical framework of the paper is presented in Figure 2. The definitions of circular economy processes and simulation modelling tools and methods are the subject of Section 2. A research gap is described in the same section as the summary of literature analysis. The scientific and practical challenges are presented in Section 3 to define the research gap in detail, while the main concept of the new simulation modelling method is presented in Section 4 to fill this gap. A presentation of a practical example of the use of the new approach in modeling the remanufacturing process and a discussion of the results in the context of existing solutions based on DES (Discrete Event Simulation) can be found in Section 5. The key conclusions and future directions are presented in the final section (Section 6).

---

**Introduction**
   justification for the topic, purpose of the article, highlights

↓

**2 Definitions (Literature background)**
of processes implemented in the circular economy – CEP
of tools and methods used in simulation
summary: description of research gap

↓

**3 Challenges**
Scientific: Factory 3D layout, Relationship, Operations, Token, Lean
Practical

↓

**4 New concept**
Application of Lean
Structuring a remanufacturing company
Structuring remanufacturing processes

↓

**5 Results**
Example
Discussion

↓

**6 Conclusions—Problem Solved vs Digital Twin**

**Figure 2.** Logical framework of the article.

## 2. Definitions: Literature Background

The circular economy prioritizes value retention, aiming to extend the lifespan of resources for as long as possible. This restorative and regenerative model, by design, fosters a new supply chain paradigm: the circular supply chain [8]. These circular supply chain systems (CLSCs) integrate material recovery processes like remanufacturing, recycling, repair, and reuse [9,10]. As Golinska-Dawson et al. [4] point out, remanufacturing has become a cornerstone of this emerging circular economy. The concept of remanufacturing suffers from ambiguity due to the presence of several, sometimes conflicting, definitions in the literature [11–14]. A very good definition is formed by Lund [15,16].

In order to standardize and introduce clarity regarding the terms used, the following concepts have been defined. They were divided into two groups:

- Definitions of processes implemented in the circular economy (CEP—circular economy processes);
- Definitions of tools and methods used in simulation.

### 2.1. Definitions of Processes Implemented in the Circular Economy—CEP

Repair: Refers to restoring a product to working order through limited dismantling and component replacement. While functionality is restored, the quality standards of the repaired products may not always match those of entirely new products.

Refurbished: Products undergo a comprehensive renewal process, including disassembly down to the module level. Damaged or outdated parts are replaced, and the product is treated to achieve a specific quality standard. This quality is typically high, but may not be identical to the entirely new products.

Remanufacturing: This industrial process involves a complete overhaul of a product. Products are disassembled to individual parts, meticulously inspected, and then undergo various actions: replacement with entirely new parts, the reprocessing of usable parts to meet original specifications, and the restoration of worn components. Through this in-depth process, remanufactured products achieve quality standards that are equivalent to, or even exceed, those of new products. However, remanufacturing typically requires more effort and resources compared to refurbishment.

Cannibalization: Involves the selective disassembly of returned products to recover usable parts. These parts can then be used for various purposes, such as repairing other products or creating entirely new ones. The quality of the cannibalized parts ultimately determines their suitability for different applications.

Recycling: This process breaks down used products into their base materials through various separation techniques. These recovered materials can then be used to create entirely new products. Recycling typically results in a loss of the original product's form and functionality.

Disposal: This is the final destination for returned products that are too costly or technically impossible to repair, refurbish, or remanufacture. Additionally, disposal might be necessary when there is no established market for the recycled materials.

These processes are carried out in the factory. A factory is defined as a form of industrial production organization based on the work of machines [17]. The establishment of factories was associated with the transition from handicraft production to mechanized production based on technical inventions. This was the result of the Industrial Revolution. Advancements in technology, including computerization and the progressive integration of processes, are revolutionizing how factories operate. These innovations are transforming both the organization of manufacturing processes themselves and the way factories are managed. However, the fundamental purpose of a factory remains the same: to produce specific products in a defined manner. To achieve this core function, a well-coordinated production and logistics system is required. This system encompasses key activities like:

- Material flow conversion—engineering approach—technology;
- Control of the flow of materials—situational and managerial—operational approach.

In this context, the concept of a factory is justified, because in the remanufacturing process we have mechanized operations such as disassembly, assembly, etc. However, intelligent remanufacturing is a relatively new and emerging research area in the literature. It has been described in various categories such as:

- Remanufacturing 4.0 [18];
- Remanufacturing enabled by I4.0 [19];
- Data-driven remanufacturing [20];
- Digital remanufacturing [21];
- Intelligent remanufacturing [19].

While some argue that Industry 4.0 (I4.0) paradigms are not directly applicable to remanufacturing due to its inherent complexity (compared to traditional manufacturing) [22], others believe its potential remains significant [23]. A well-defined research program that considers the technological, business, economic, social, and environmental aspects of "smart remanufacturing" (as suggested in the work presented in [22]) could bridge this gap and unlock the transformative potential of I4.0 for the remanufacturing industry.

*2.2. Definitions of Tools and Methods Used in Simulation Modeling*

Simulation modeling involves creating a digital replica of a system from the real world. This replica or model reflects the key components and their behaviors that are relevant to the specific problem you are trying to solve. The key here is to make the structure of the model as close as possible to that of the real system. In other words, every important element in the real system corresponds to an element in the model. When building a simulation model, you define how each element works and how it interacts with each other. To analyze the system, you then conduct experiments within the simulation model. The process that takes place during the experiment is similar to the process that occurs in a real object, so the study of the object through its simulation model involves studying the characteristics of the process occurring during the experiment. Simulation modeling has emerged as a powerful tool in production and remanufacturing research, offering valuable insights into overcoming challenges, particularly those related to uncertainty [24].

DT—Digital Twin— is a virtual representation of a physical system, i.e., an environment, process, object with its elements, features and functionalities, the level of reflection of which enables the simulation and evaluation of alternative scenarios of the future without any loss of quality of simulated events compared to real events in terms of the key parameters for the decision-maker [25].

DES—Discrete Event Simulation—is a modeling technique in which any changes in the simulation model are represented by events (nodes) that occur at the time when certain conditions occur. Such models are typically represented by event graphs with nodes and dependencies between them (arrows). Each event occurs at a specific point in time and represents a change in state in the system. It is assumed that there will be no change in the system between events. In this way, the simulation time can directly jump to the time of the next event [26].

ABS—Agent-Based Simulation—is a powerful simulation technique that models complex systems by using autonomous components called agents [27]. These agents interact with each other and their environment, influencing the overall system's behavior. Each agent's behavior determines its role, how it interacts with others, how it responds to messages, and even if it can adapt its behavior based on the environment [28].

Structuring—isolating the structure of something or giving something some structure (structuralize, structure) [29]. This concept is related to the concept of Structure (Latin: structura—"construction, way of building"), which means the arrangement of the components of a given system and a set of relations (mutual connections) between these elements, characteristic of this system; the way in which the parts of a certain whole are related to each other. Structure is what gives the whole unity without the need for analysis and synthesis, it is a constant element of the organized whole, recognizable in spite of the

changes that take place in this whole. In a structure, the individual elements mean nothing or little in themselves. Only participation in the whole gives them meaning.

### 2.3. Definition of Research Gap

Despite extensive research on remanufacturing itself [30], the application of simulation modeling to facilitate these processes remains a relatively unexplored area. A search in SCOPUS using "simulat" OR "modelling" AND "remanufact" within article titles yielded only 56 results. In contrast, Lee and Kwak [30] identified 369 articles on remanufacturing in a single journal, with a focus on 'supply chain', 'environmental', and 'sustainability' aspects. This comparison highlights a clear gap in research on leveraging simulation for remanufacturing. Similar to traditional manufacturing, simulation modeling offers significant advantages for remanufacturing. It provides valuable insights into production processes, helping to predict behavior on the shop floor. This allows manufacturers to proactively suggest solutions based on real-time analysis, optimizing remanufacturing operations. However, remanufacturing presents unique challenges and uncertainties compared to traditional manufacturing. These include price fluctuations, stochastic demand, and core challenges related to used products or parts. The uncertain quality of returned items, time constraints, and ensuring a consistent quality throughout the process are all major hurdles [31]. The European Remanufacturing Network (ERN) further highlights the lack of accurate, timely, and consistent product knowledge as a critical industry-wide issue based on their survey of 188 European remanufacturers.

The scientific challenges result primarily from the comparison of the expectations and challenges of Industry 4.0 in relation to simulation methods with the methods and tools currently available, in terms of methods and tools offered on the market and described in the literature.

A wide range of mature and industry-leading simulation software is available from numerous manufacturers today. The comparisons of simulation systems are carried out on an ongoing basis by various organizations, for example, the DES simulator ranking can be found in the work presented in [32]. The leading ones are (alphabetically): Anylogic, Arena, AutoMod, FlexSim, OPS, Plant Simulation, Promodel, Simio, and Witness.

Each of the manufacturers proposes a methodology for creating simulation models that comply with the principles defined in the work presented in [33], consisting of:

- Defining the problem and defining the purpose of the simulation;
- Operational characteristics of the simulation model;
- Construction of the model;
- Designing experiments;
- Analysis of the results of experiments;
- Validation and practical verification.

Many simulation software tools utilize an object-oriented approach. In this approach, users construct models by selecting objects (pre-built components) from a library and arranging them on a virtual workspace using a mouse [34,35]. While this typically involves building models from the ground up, these software objects are becoming increasingly sophisticated, offering more intelligent behavior. Automatic model generation is an active area of research [36], but commercially available solutions are still in development, meaning that they are not yet widely adopted as a standard workflow.

Most simulation software falls into the category of general purpose tools. This versatility allows them to be applied to a wide range of problems across different industries. However, this flexibility also presents a challenge for manufacturers, who need to cater to a broad user base. To address specific industry needs, some companies offer specialized versions of their software. For example, FlexSim [37] provides industry-tailored solutions like FlexSim Healthcare and FlexTerm for container terminals. Similarly, Haulsim focuses on open-pit mine operations. These specialized programs leverage the core FlexSim simulation engine. Other leading manufacturers, such as Anylogic, Simio, and Promodel, offer a different approach. They provide libraries of pre-built objects and processes designed for

specific applications. For instance, Anylogic offers dedicated libraries to address various industry needs [38]:

- Process Modeling Library for general business processes or workflows;
- Fluid Library to simulate the transportation of bulk cargo and liquids in industries such as mining or oil and gas;
- Rail Library, for rail transport terminals, and trans-shipment yards;
- Material Handling Library for production and storage processes;
- Road Traffic Library for the movement of cars, trucks, and buses on roads, parking lots and factory areas;
- Pedestrian Library for pedestrian traffic at airports, stadiums, railway stations, or shopping malls.

While libraries offer a valuable resource for building models, the focus of development remains on expanding the object library itself, not on automating the entire model creation process. This means users still need specialized skills in object modeling, system analysis, and potentially some programming knowledge. Additionally, even though users can save their own custom objects for reuse, this typically requires more advanced IT skills. It can be assumed that these tools are characterized by the following features.

The basic concept is "Abstract"—that is, the use of abstract concepts and structures with the aim of achieving an effect in the model similar to that in reality. This requires knowledge of computer science, modeling, programming—usually new skills for engineers and factory personnel. This is a low level of modeling that also requires knowledge of the so-called interfaces: objects, programming tools, programming languages, scripts, and tables. In practice, using simulation software often necessitates hiring a dedicated modeling analyst to build factory simulations. For production engineers, logisticians, or Lean specialists, this translates to acquiring new skills and experience, which can be a lengthy process. Furthermore, this specialized knowledge may not be directly applicable to their day-to-day jobs. It is important to note that simulation software often relies on abstraction tools like State Charts or Process Flow diagrams. These tools require users to create models using abstract elements, such as "tokens," that represent real-world objects or processes. While these tools can be powerful, the need for abstraction adds another layer of complexity to the modeling process.

Another challenge is the need to translate real-world processes into abstract concepts within the simulation software. These abstractions, like "tokens," are unfamiliar to most factory personnel and engineers. This terminology gap between the simulation software and the factory floor can be a significant hurdle. Furthermore, mastering a simulation program's interface adds another layer of complexity. These interfaces often involve a diverse mix of objects, tables, scripts, and functions—hundreds in some cases. While this versatility allows for broad application, it also comes at the cost of a lengthy learning curve. The entire process, from problem definition and model development to testing, data collection, and performance measurement, can be quite time-consuming.

The expertise required to effectively use simulation software adds to the overall cost of implementation. Modelers need a broad skillset encompassing simulation theory, statistics, software proficiency, and potentially programming knowledge. This translates to significant expenses beyond just the software itself. Training, stakeholder time investment, data preparation, and potentially consultant fees for model building all contribute to the cost. Additionally, the complexity of simulation projects introduces a high risk of scope creep and delays, which can further inflate costs.

## 3. Scientific and Practical Challenges

### 3.1. Scientific Challenges

The primary scientific challenge for simulation modeling in remanufacturing lies in bridging the gap between real-world remanufacturing processes and the virtual world of simulation. Ideally, we should be able to seamlessly translate a company's remanufacturing activities (repair, restoration, remanufacturing, cannibalization, recycling, utilization)

into a simulation model that reflects the factory layout, relationships, and other key aspects, just as real-world factories are designed and built. To achieve this, we can define a set of scientific challenges or requirements for simulation modeling, specifically tailored to remanufacturing processes. These challenges can be categorized into different areas, including:

- Factory layout—3D;
- Relationship;
- Operation;
- Token;
- Lean.

### 3.1.1. Factory Layout 3D

Factory layout refers to the arrangement of all physical elements within a production facility. Traditionally, layouts are created as 2D plans (often in .dwg format using AutoCAD) viewed from above. These macro-level layouts focus on ensuring a smooth flow of materials, often utilizing linear or nested workstation arrangements to minimize transport times between cooperating stations. Micro-layouts, on the other hand, focus on the detailed arrangement of individual workstations or designated work areas. This includes the machinery, operator workspace, and their movement patterns within the area. The key challenge presented in this article is the adoption of "factory topography." This concept views the factory layout as a three-dimensional configuration, considering both the physical layout (2D projection) and the presence and location of objects and key points. These points could include stations (work tables, machines, conveyors, etc.), communication routes, passage points, and logistic train stops. This article proposes using workstations as the fundamental object in this system. Each workstation would have a user-assigned name, and its components would have standardized designations. This systematic approach allows for the unambiguous identification (addressing) of any element within the factory. Furthermore, creating the layout in 3D enables true addressing that considers the Z-axis (height). This opens the door for using advanced technologies like Virtual Reality (VR) and Augmented Reality (AR) to visualize and interact with the factory layout in a more immersive way.

### 3.1.2. Relationship

The unambiguous identification of factory components is crucial for accurately capturing the relationships between them. A key challenge lies in leveraging the PFEP (Plan For Every Part) database as the primary integration point for these relationships. The PFEP database already holds detailed information about parts used in manufacturing. By utilizing this existing data source, we can potentially streamline the process of defining and recording factory component relationships. The predefined locations should be related to the bins in which the parts flow in such a way that there can be only one type of parts bins at a given location according to the Lean rules set out in the work presented in [39]. Containers, on the other hand, are linked to the parts that flow in them through the PFEP database. It is a database that supports the correct and controlled implementation of the material distribution system inside the factory. The implementation of a PFEP-based system requires knowledge of each of the components and materials used in the company for production or disassembly, knowledge of the position at which they are used, what is the demand for them, how and where they are purchased, how they are delivered, where they are stored, how they are packaged, etc. Properly established, filled with reliable information about the parts used on the shop floor, and properly managed, PFEP enables [39]:

- Rapid start of work on the development of a lean material flow system and its further development;
- Gathering up-to-date data for all parts used in production in one central place;
- Sorting data by any category, which improves, e.g., the design of a supermarket of purchased parts;

- Quick access to supplier data and a response to possible supply disruptions;
- Extending lean material flow to suppliers and customers.

PFEP is built on the basis of the structure of the product and is linked to the manufacturing logic diagram, which describes what happens to each detail during manufacturing and remanufacturing. Thus, the manufacturing and remanufacturing process of a given product is related to the structure of that product and to the PFEP assigned to the product. In other words, each product has its own PFEP. The changeover of the station to the production of another product is related to the switch to the PFEP assigned to that product. The scientific challenge is to understand the essence of PFEP and to build it into a simulation application (simulator), which will enable the automation of the construction of the simulation model.

### 3.1.3. Operations

Simulating operator movements is another crucial challenge. In factories, workflow relies not just on human operators but also on mobile entities like robots, manipulators, forklifts, and automated vehicles (AGVs and AMRs) that transport parts and containers. These entities follow pre-defined routes consisting of tasks or work cycles. The challenge lies in translating this concept of operator routes into the simulation program. This requires developing a high-level language specifically designed to describe these routes in a way that mirrors how they are created in real factories (similar to publication [40]). Ideally, this language would be user-friendly and allow for:

- Conflict resolution: Built-in mechanisms to handle situations where multiple entities need access to shared resources, ensuring smooth operation without collisions;
- Value-added analysis: Assigning attributes to actions within the route to facilitate value-added analyses and the creation of Yamazumi diagrams (tools for analyzing production efficiency).

### 3.1.4. Token

Current simulation software relies on tokens as abstract objects that move through a flowchart to represent processes. While this approach helps to visualize the flow and describe reality at a high level, it can be challenging for factory engineers to understand due to its abstraction. The scientific challenge here is to develop alternative process descriptions that are more intuitive and align better with the existing knowledge of factory personnel. This could involve leveraging concepts from Lean manufacturing and the multimodal approach. Imagine the factory floor as a network of locations where containers holding parts and finished products move around. We could then define the token as the container itself, with local cyclical processes handling activities within each location. Alternatively, we could view the flow as happening between locations, with workstations or production cells performing operations on the containers as they move. This shift in perspective would provide a more factory-centric view of the processes and flows taking place.

### 3.1.5. Lean

- Lean manufacturing offers a well-established methodology for designing and managing a factory's internal logistics system [39,41]. A key scientific challenge is to integrate these Lean concepts into simulation programs. This would significantly speed up model building and create a more familiar environment for production engineers, logisticians, and Lean specialists. Here's how Lean principles could be embedded within the simulation program:
- PFEP, which has already been mentioned;
- Automated Value-Added Analysis (VA): This would require a high-level language for describing routes, allowing the program to automatically identify and analyze value-added activities within the model;

- Yamazumi Charts: A high-level language and cyclical route definition approach would also be necessary to support the creation and utilization of Yamazumi charts, which help to balance workload across workstations;
- Andon and Kanban Mechanisms: These mechanisms, respectively, for signaling issues and controlling material flow based on location occupancy, could be integrated to provide real-time feedback and optimization within the simulation;
- MilkRun Delivery Management: The program could handle the generation and management of MilkRun deliveries, a logistical approach that utilizes designated routes for efficient parts transportation.
- By incorporating these Lean principles, simulation programs can become more user-friendly and effective for production personnel familiar with Lean practices.

3.1.6. Summary

- Overcoming these scientific challenges (summarized in Table 1) would significantly improve the accessibility and usability of simulation technology for factory engineers. Here is how:
- Reduced Workload: By automating model generation through built-in relationships and leveraging familiar Lean terminology, engineers can spend less time on model creation and more time on analysis and optimization;
- Automated Analysis: The program could automatically generate value-added charts, Yamazumi charts, and ergonomic data (distances traveled, workload analysis) to streamline the process and provide valuable insights;
- Improved Decision-Making: Separating planning and execution phases would provide a structured approach to decision-making, allowing engineers to develop models based on reference processes and logistics navigators;
- Enhanced Visualization: 3D visualization and integration with VR and AR technologies would create a more immersive and intuitive experience for analyzing and interacting with factory simulations.

**Table 1.** Summary of scientific challenges in the form of a set of features and their description for the CEP simulation modeling method.

| Characteristic | Scientific Challenge |
|---|---|
| 3D factory layout | Factory topography refers to a three-dimensional (3D) representation of a factory layout. It goes beyond a simple top-down view (2D layout) by considering not only the physical shape of the factory floor but also the presence and arrangement of key elements within it. These key elements can include: Stations (work areas containing machines, tables, conveyors, etc.), Machines and other equipment, Storage locations, Passageways and traffic areas, Key process points. By incorporating this 3D perspective, factory topography provides a more comprehensive understanding of the factory environment and its spatial relationships. |
| Relations | The system of relations existing in the factory, taking into account the PFEP—Plan for Every Part |
| Operations | Operator Movement Simulation and High-Level Route Language: This combined title captures both aspects—simulating operator movements and the development of a language to describe routes. |
| "token" | This approach views the token as a physical container carrying parts or finished products. The actual work is then performed by local cyclical processes happening within specific locations in the factory. |
| Lean | Developing a way to use the achievements of Lean Manufacturing in a simulation program |

*3.2. Practical Challenges*

The key challenge lies in translating the scientific advancements identified in the previous chapter into practical applications through the development of a user-friendly simulation tool. This tool should effectively address the real-world challenges faced in remanufacturing. Practical challenges can be defined as a set of features that should be met by such a tool dedicated to modeling processes implemented in the circular economy. See Table 2, where the specification of the features for a simulation program is listed.

**Table 2.** Summary of practical challenges in the form of a set of features and their description for the CEP simulation modeling method.

| Characteristic | Practical Challenge |
|---|---|
| Purpose | Dedicated to the CEP |
| Types of issues | Design and analysis of systems implemented in the processes: Repair, Renewal, Remanufacturing, Cannibalization, Recycling, Utilization |
| Frequency of use | Reusable |
| User | Production and Engineering, Operations, and Logistics |
| Limitations | Remanufacturing Focus |
| Basic concept | Focused, extension of the user's (engineer's) work environment |
| Terminology | Remanufacturing, logistics, Lean; mainly daily deadlines for engineers and factory workers carrying out remanufacturing processes |
| Interface | Tables and Actions |
| Time for results | Rapid Results (Known System Configuration) |
| Resources required | Minimal Resources: Existing Production Staff with Lean Expertise |
| Knowledge/training required | Low Barrier to Entry: Focus on User Interface and Basic Tools |

## 4. The Concept of the Simulation Modeling Method

The concept of the simulation modeling method for the CEP project is based primarily on the coherent integration of the methods offered in the Lean Manufacturing concept into the simulation program. The concept being developed takes into account and responds to previously defined scientific and practical challenges. The following way of formulating this concept was adopted, starting from the Lean Manufacturing methodology (Section 4.1) (by defining its most important components to be built into the simulation program), developing the concept of structuring (i.e., defining the arrangement of the components of a given system and the set of relations (interconnections) between these elements, characteristic of this system, i.e., the way in which parts of a given whole are related to each other) both at the level of factory resources (Section 4.2) and at the level of processes (Section 4.3) implemented in this factory.

### 4.1. The Application of Lean Principles in Simulation Modeling

The concept of using Lean Manufacturing methods has already been described by the author of this report in the publication presented in [40]. It is proposed to use this approach in the developed simulation modeling method. It is based primarily on defining a high-level language, enabling the description of work instructions that are used in industrial reality, in a simulation program. A feature of this language is the mapping of work carried out in reality in the form of routes in exactly the same way—1:1. They should take into account those properties that are naturally used by the employee, i.e., his orientation in space, operation in the check and wait mode, i.e., first check whether the conditions for the task implementation are met, if not, wait until they are met. This mechanism should include also solving the problem of access to shared resources, e.g., when two or more employees try to gain access to a common resource, e.g., downloading a part, tool, container. Meeting these requirements will allow instructions to be assigned the attributes of work that adds value (VA—Value Added), work that does not add value (NVA—Non Value Added), but which must be performed, e.g., quality control, and work that does not add value but it is attackable, i.e., we want to minimize and eliminate it (NVAA—Non Value Added Attackable). Embedding such a defined language into the simulation program enables the construction of operator routes identical to those in reality (1:1) and the automatic performance of VA (Value Added Analysis) analyzes and the automatic generation of operator workload diagrams, the so-called Yamazumi diagrams. Incorporating other

Lean tools into the simulation program, such as PFEP, Andon, and Kanban, requires the development of a factory structuring concept and process structuring on this basis.

Figure 3 shows the methodology for the design/redesign/modeling of dismantling/ recycling systems for CEP. The main data preparation/collection phase is divided into four main stages:

- Process data;
- Factory/system topography;
- Connections for the intralogistics system based on PFEP;
- Connections for AGV/AMR systems.

Depending on the scale of the project, it is possible to use:

- Excel spreadsheets for a small scale project;
- Simulations for situational solutions;
- Comprehensive/holistic digital Lean twin of the factory (process), using simulation, for system solutions.

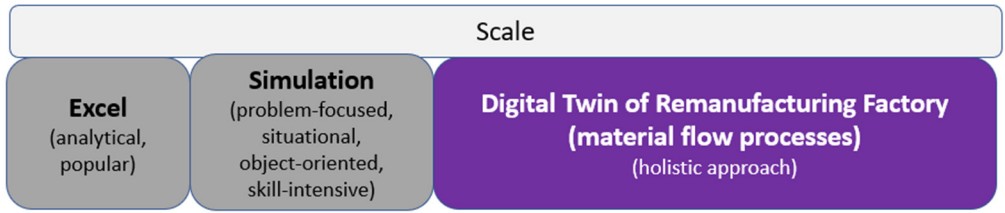

**Figure 3.** Methodology for designing/redesigning a remanufacturing system.

The introduction of the Lean Factory Digital Twin concept requires structuring the factory and the processes implemented in this factory. This is the subject of the following sections.

### 4.2. The Concept of Structuring a Remanufacturing Company

For factory structuring according to purposes of the CEP project, it is proposed to define:

- Logical relationships;
- Spatial relations;
- The factory facility identification system.

Logical relations are a system of defining relations between resources that constitute the material base of the factory implementing CEP. The concept of relationship definition is presented in Figure 4. A scheme used in the description of relational databases was adopted, where relationships between databases are shown in the form of aggregation relationships (solid arrows in Figure 4) and dependencies (dashed arrows in Figure 4).

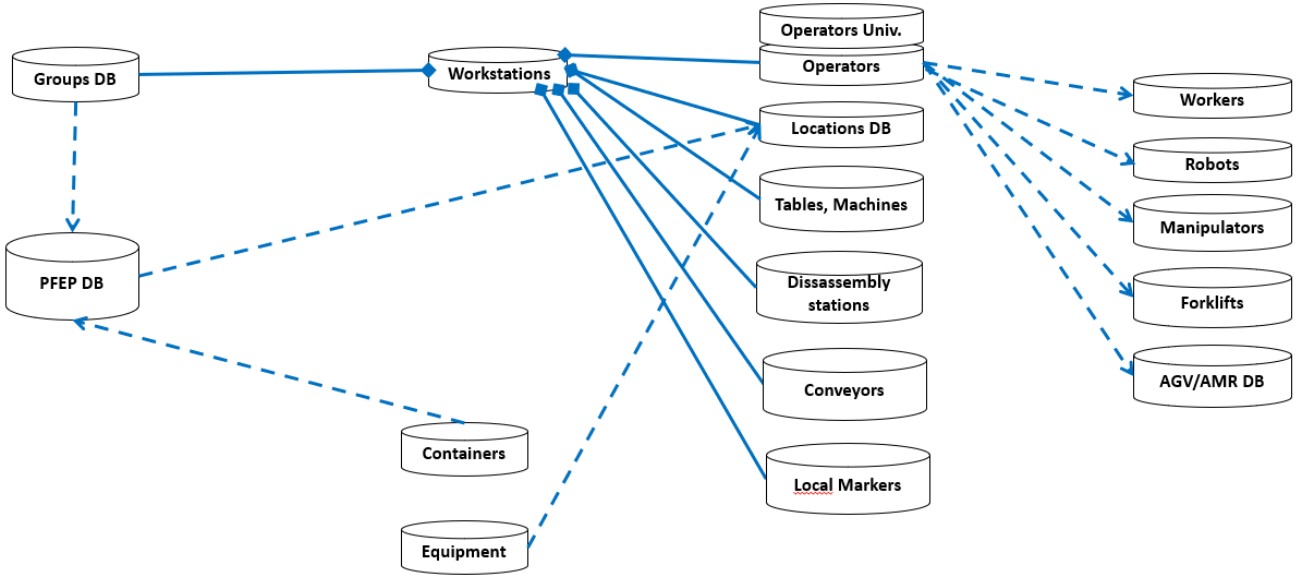

**Figure 4.** The concept of the factory's logical relations system for the CEP project.

List of databases:

Groups DB—table of job groups. The group includes the workstations that make up the table—Workstations. It includes:

- Operators (working independently)—Operators table;
- Universal operators (forming a team)—Operators Univ. table;
- Locations—distinct areas within the system where containers with parts and products are located, along with operator and logistics access points—Locations DB table;
- Work tables, machines—Tables, Machines table;
- Disassembly stations—Disassembly stations table;
- Conveyors—Conveyors board;
- Markers on the floor symbolizing places of stopping and approaching—Local markers.

A given group of workstations is related to the PFEP by the PFEP table containing a list of these plans. Each PFEP uses the Containers table, containing a list of containers with their description. The PFEP table is linked to the Locations DB table through the part number because a given location can contain containers with only one type of part—a Lean Manufacturing requirement. The location may also contain equipment elements, hence the connection of the Locations DB table with the Equipment table containing a list of equipment and its description. Spatial relations concern x, y, z coordinates in 3D space. This means that each element listed in the tables, creating a system of logical relations, has its own geometric coordinates and size. The developed concept adopts the logic of thinking proposed by Lean regarding the layout of the factory. According to Lean, the layout is the arrangement of machines in the production hall, a bird's eye view of the entire area or part of the room with the location of all elements. In the context of production management, the area shown on the layout is the production hall, and the most important elements from the point of view of Lean are the machines and workstations.

Lean manufacturing emphasizes two key layout concepts:

- Macro Layout: This refers to the overall arrangement of all elements within the production facility. Lean principles aim to create a continuous flow by strategically positioning workstations. Linear or nested layouts are preferred to minimize transport distances between workstations, ultimately reducing the overall production time;
- Micro Layout: This focuses on the detailed arrangement of individual workstations or designated work areas. It considers all the necessary equipment, the operator's

workspace, and their movement patterns within that space. The goal is to optimize the workstation environment for efficient task completion.

As part of the developed concept, it is proposed that the stations are defined in one global system of the entire factory, while the elements constituting the station are defined in the local system of a given station. This means that each site has its own local layout Figure 5.

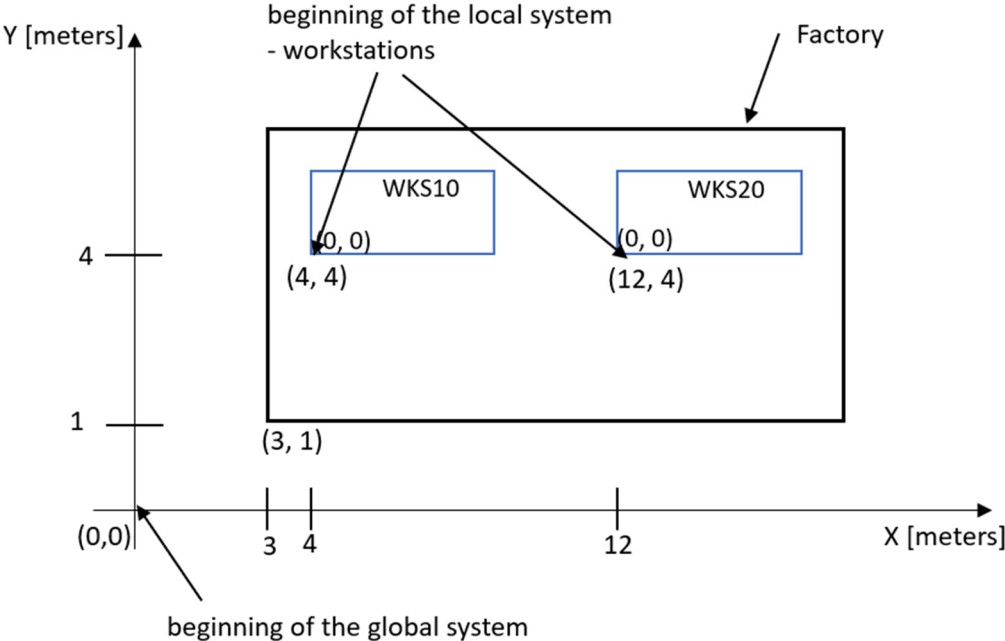

**Figure 5.** Presentation of the proposed global system and local systems.

This also means that the objects inside the station are related to each other and moving the station (in the global system) automatically moves all its components. In this way, when talking about the global layout, we mean the layout on a macro scale, and when talking about the local layout of the site, we mean the layout on the micro scale.

This way of thinking determines the third element of factory structuring—the factory facility identification system. Its essence is for each object constituting the factory to be clearly identified. Referring to the global/local (macro/micro) system, it is proposed to use the following method of identifying the factory elements:

Global Names—any name of the position and group of positions

- WKS_xx workstation—any name;

Local Names inside the site all are reserved;

- P_xx—location designation—the place where the container, in which parts and assemblies are transported, is placed;
- N_xx—the marking of a point on the ground—the point of the operator's approach to the corresponding location (same xx number);
- I_xx—the marking of a point on the ground—the logistician's approach point to the corresponding location (same number xx);
- Diss_xx—the designation of the disassembly location;
- Op_xx—the operator designation inside the WKS;
- Ou_xx—the designation of the universal operator inside the WKS—universal operators are assigned universal cycles. WKS contains a decision-making center that assigns cycles to Ou_xx;
- WT_xx— the designation of the work table, machine;
- G_xx—the designation of a point in the local system of the workstation—e.g., it is used to indicate the point of approach to the assembly station;

- Cnv_xx—conveyor designation.

### 4.3. The Concept of Structuring Remanufacturing Processes

The concept of structuring remanufacturing processes is based on the idea of high-level script language describing agent behavior in agent-based modeling. It is proposed, on the one hand, to use this type of scripting language with the attribute mechanism presented in Section 4.1 and to introduce a clear division into the process execution level and the process control level. The implementation of this concept will allow structuring remanufacturing processes. The essence of this concept is to enable the transition from Lean to simulation in the first step and from simulation to Lean in the second step. This situation is presented Figure 6.

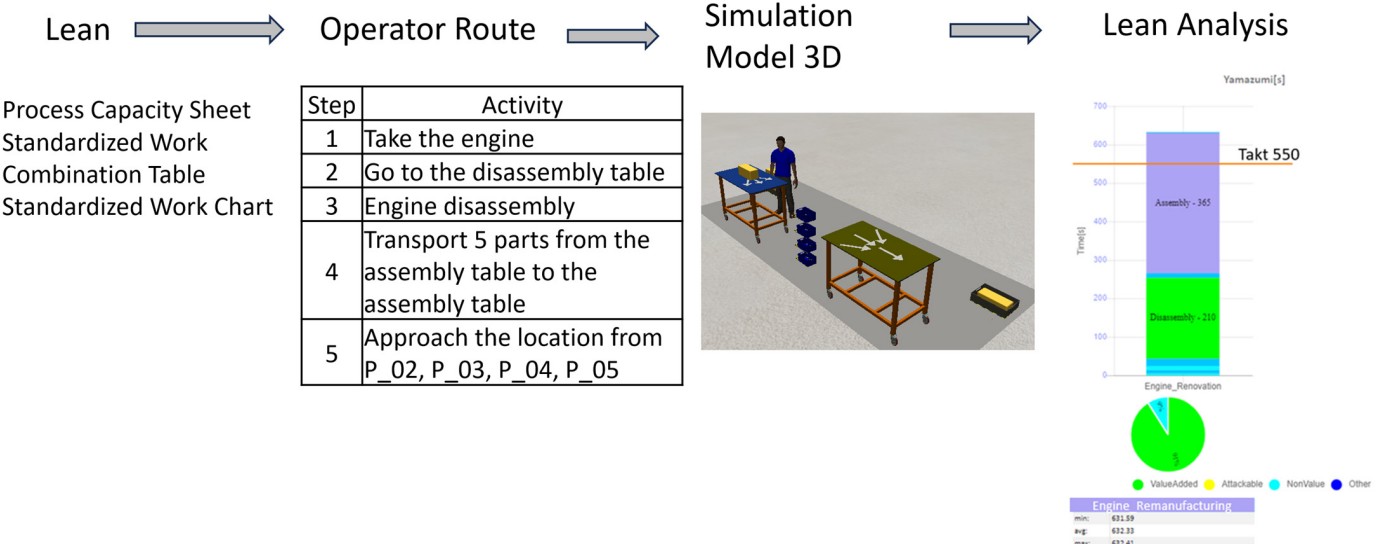

**Figure 6.** Transition from Lean (standardized work) to simulation (operator routes) and to Lean (Analysis: VA, Cycles, Balancing the workload of operators/workstations).

The transition from Lean to simulation involves the use of Lean tools.

Standardized work defines the most efficient way to complete tasks within a production process. It is built on three key elements:

- Takt time: this is the rate at which units need to be produced to meet customer demand;
- The exact sequence of work (cycle): this refers to the specific order and timing of tasks an operator performs to complete a unit;
- Standard inventory: the necessary inventory (including units in machines) to ensure the process runs smoothly without interruptions.

Standardized work serves as a foundation for continuous improvement through kaizen. Once established and displayed at workstations, it becomes a documented reference for all shifts, reducing process variability. This streamlined approach simplifies training new operators, minimizes the risk of injuries and overload, and establishes a clear benchmark for ongoing improvement activities.

Standardized work is typically documented in three key formats. These formats are used by engineers and frontline managers to design and refine the process. Operators also leverage them to identify opportunities for improvement within their own tasks.

Process performance sheet—identifying bottlenecks and optimizing production. This form helps to analyze the performance of a production cell (a connected set of processes) by comparing actual performance against ideal output. It acts as a tool to pinpoint bottlenecks and areas for improvement. The sheet captures key metrics like: machine cycle times, setup and tool change times, manual labor times. By analyzing this data, you can identify

bottlenecks that hinder overall cell performance and implement strategies to eliminate them.

Standardized table of working connections—a deeper look at production flow. This form goes beyond the operator balance chart by providing a more granular breakdown of each operator's tasks. It details the combination of manual time: the time an operator spends directly working on a unit; transition time: the time spent moving units or materials between tasks; and machine processing time: the time a machine takes to complete a specific operation on a unit. The completed table offers a more precise picture of the interactions between operators and machines within the production sequence. This level of detail allows for recalculating an operator's workload as the takt time fluctuates.

The standardized work pattern provides a visual guide for efficient production. This chart serves as a visual roadmap for each operator's tasks within the production process. It captures the three key elements of standardized work: takt time and cycle time and work sequence. The standard work materials are the necessary inventory. The benefits of this pattern are displayed directly at workstations. The chart promotes transparency and facilitates continuous improvement (kaizen); it is dynamic and adaptable—the chart can be reviewed and updated as the workplace layout or process efficiency improves.

The description of standard work is proposed to be implemented in a scripting language that allows for assigning added work attributes (see Section 4.1) to the performed operations. This description is proposed to be presented in the form of tables (Table 3) with a column structure:

- Where—factory object identifier;
- Activity—a high-level language statement defining the work performed on the object defined in Where;
- Param—a high-level language numeric parameter;
- Description—description of the work performed, additional parameters.

**Table 3.** A table describing work in a scripting language.

| Where | Activity | Param | Description |
|---|---|---|---|
| Diss_01 | Unload | 1 | Unload Item on Table |
| Diss_02 | Work | 30 | Disassemble for 30 s |

Due to the fact that the purpose of introducing standardized work is to ensure its repeatability, the table would represent the so-called Duty cycle. Many cycles would be recorded in many tables, which means that the employee would carry out the work described in the job instructions (i.e., he would carry out the cycle described in the table), changing the employee's task by moving to another job would be associated with a change in the work cycle. Changing a task means managing the work of operators (controlling the implementation of work cycles). The process is structured by dividing it into two distinct levels: executive and control. This separation is illustrated in Figure 7.

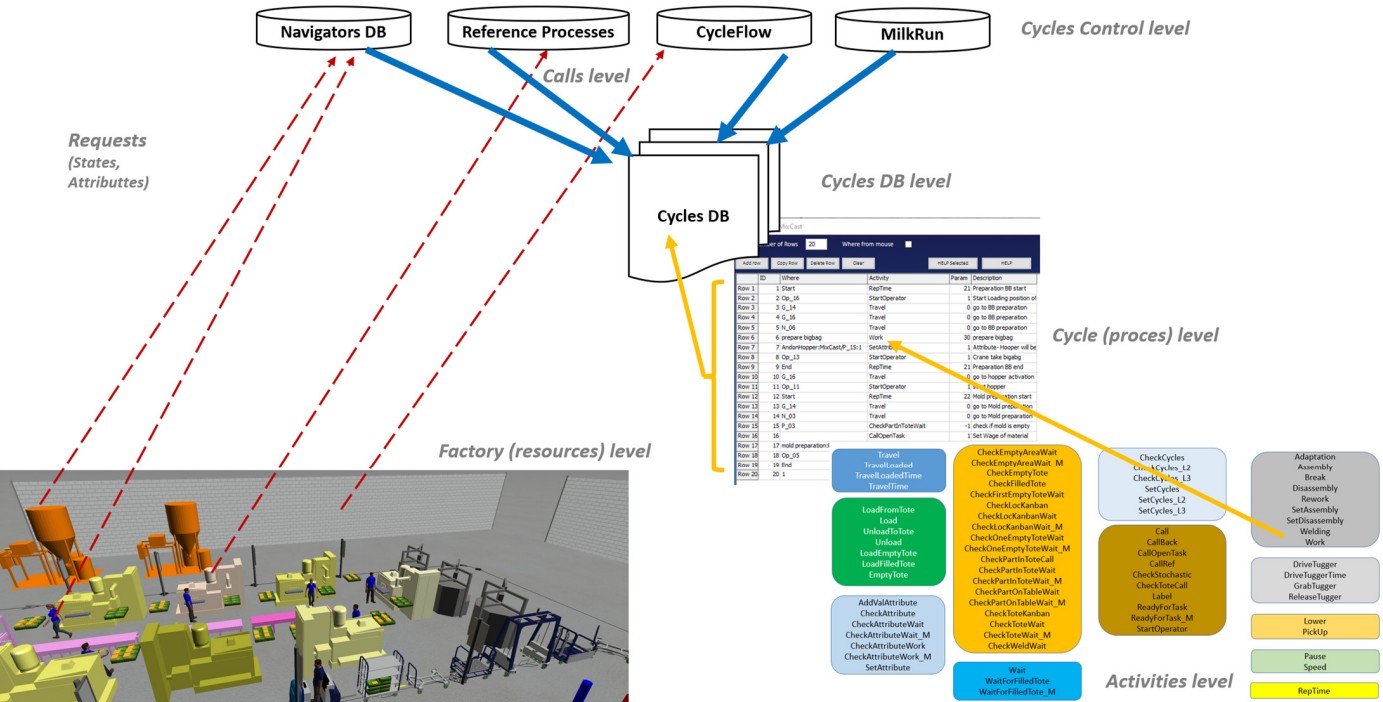

**Figure 7.** Process structuring with distinction of levels: activity, cycles and control.

## 5. Results and Discussion

### 5.1. Results of the Practical Example

The proposed simulation modeling concept aligns with defined scientific and practical needs. It facilitates the rapid development of simulation models, a cornerstone of digital twin technology. This approach embeds synchronization and access control for shared resources directly within the high-level language instructions. This eliminates conflicts even when multiple operators run the same instructions simultaneously. The solution leverages two key aspects:

- Linking the high-level language with the 3D factory layout and its notation. This allows the instructions to understand the physical environment and resource locations;
- Building a mechanism for synchronized access based on Lean principles (instruction evaluation). This ensures efficient resource allocation.

Workers can autonomously switch cycles using a decision tree mechanism embedded in the instructions. Alternatively, an external control unit, like a foreman or an industrial automation system, can manage cycle assignment and worker switching.

The new method, based on a high-level script, uses the Lean management concept and consists of the following steps:

- Prepare product related data (Bill of Materials);
- Prepare PFEP—a database that ties together the parts and the containers in which the parts are transported in the remanufacturing system. When creating the PFEP, complete the container database (dimensions, weight, shapes) and the parts database (dimensions, weight, shapes);
- Define a group of workstations where disassembly, inspection, reprocessing, reassembly, and other operations are carried out. Link them to the PFEP;
- Define the workstations within each group;
- Define locations for: containers, operators, machines, material handling equipment;
- Arrange them on the layout;
- Assign a selected/specific part to a specific location;
- Think with work cycles—identify/design work cycles;
- Write down the work cycles with a high-level language;

- Define observation, who handles them, and when;
- Define Lean value-added attributes (VA, NVA, NVAA) to work instructions. Decide how to present results;
- Define other metrics if necessary—KPIs, WIPs, throughputs.

The methodology outlined above captures the 1:1 workings of the real system. This section details the implementation of the proposed approach for remanufacturing Hand Vacuum Cleaners (HVCs). Table 4 shows the PFEP built for this product.

**Table 4.** PFEP—Plan for Every Part built for HVC (Hand Vacuum Cleaner)—PFEP_HVC.

| Part | Part Name | ID | Length | Width | Height | Capacity |
|------|-----------|-----|--------|-------|--------|----------|
| BO1 | Housing | KLT6147 | 0.50 | 0.20 | 0.20 | 5 |
| BO2 | Ball bearing | KLT3147 | 0.03 | 0.01 | 0.05 | 10 |
| BO3 | Rotor | KLT3147 | 0.07 | 0.01 | 0.05 | 24 |
| BO4 | Screw | KLT3147 | 0.01 | 0.01 | 0.05 | 25 |
| BO5 | Sensor | KLT3147 | 0.04 | 0.01 | 0.05 | 10 |
| BO6 | Spring washer | KLT3147 | 0.07 | 0.01 | 0.05 | 10 |
| BO7 | Electric cable | KLT4147 | 0.10 | 0.01 | 0.05 | 8 |
| BO8 | Glue | KLT3147 | 0.01 | 0.01 | 0.05 | 30 |
| BO9 | Nameplate | KLT3147 | 0.01 | 0.01 | 0.05 | 20 |

The first two steps of the methodology involved building the PFEP base using the BOM (Bill of Materials). In Step 3 and 4, an inventory of workstations is defined (in this case only one workstation named "Engine_Remanufacturing" was formed). These workstations are grouped together as "Line_HVC" and are linked to the corresponding PFEP, named "PFEP_HVC".

Following Step 5, the locations are defined in the simulation model. A layout of the workstation (so-called microlayout), includes machines, buffers/storage fields, and position markers on the floor for operators. The exemplary workstation consists of 6 locations, one disassembly table, one assembly table, three floor markers, and one operator (worker). The workstation is arranged on a layout, as seen in Figure 8.

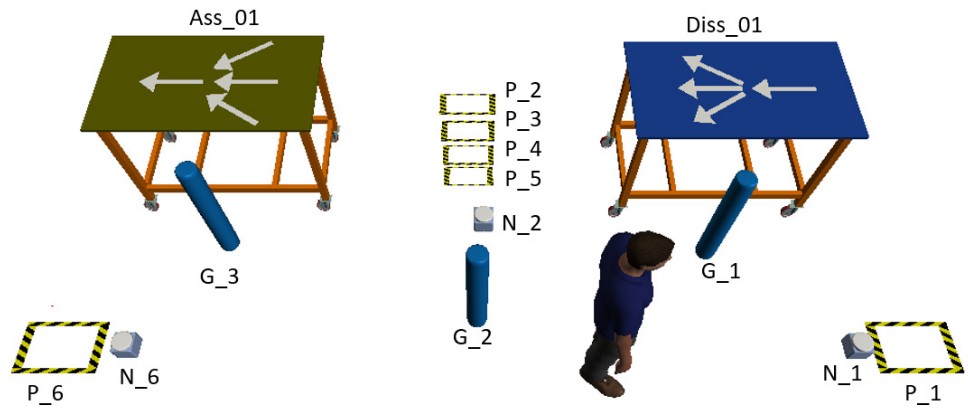

**Figure 8.** Layout of modeled remanufacturing system.

The markers P_x indicate locations, Ass_01 is the assembly table, Diss_01 is the disassembly table, G_x indicates the markers on the floor that form the passage route for the operator, which is marked by Op_01.

A reference process library is created, based on standardized work descriptions (work procedures), which determine: (1) who performs certain actions, (2) how, and (3) in what

order. The related tables include the example of reference processes written in the described high-level language. These tables are created with the participation of industrial partners in order to develop a library of processes which are relevant for the remanufacturing settings (Table 5). The main goal is to standardize activities so that they are repeatable, measurable, and effective. Thus, the construction of a library of reference processes refers to work instructions carried out at a specific the notation described in the previous work [40].

**Table 5.** The work performed by operator and the corresponding instructions of high level script language.

| Nr | Work | High Level Script Language |
|----|------|----------------------------|
| 1 | Take the engine | N_1 Travel 0<br>P_1 CheckPartInToteWait 1<br>P_1 LoadFromTote 1 |
| 2 | Go to the disassembly table | G_1 TravelLoaded 0<br>Diss_01 Unload 1 |
| 3 | Engine disassembly | Diss_01 Disassembly 210 |
| 4 | Transport 5 parts from the assembly table to the assembly table | Diss_01 Load 5<br>G_2 TravelLoaded 0<br>G_3 TravelLoaded 0 |
| 5 | Take 4 parts from locations P_02, P_03, P_04 and P_05 | G_2 TravelLoaded 0<br>N_5 TravelLoaded 0<br>P_2 CheckPartInToteWait 1<br>P_2 LoadFromTote 1<br>P_3 CheckPartInToteWait 1<br>P_3 LoadFromTote 1<br>P_4 CheckPartInToteWait 1<br>P_4 LoadFromTote 1<br>P_5 CheckPartInToteWait 1<br>P_5 LoadFromTote 1<br>G_2 TravelLoaded 0<br>G_3 TravelLoaded 0 |
| 6 | Place 4 parts on the assembly table | Ass_01 Unload 4 |
| 7 | Installation of a new engine | Work 30 |
| 8 | Transport the assembled engine | Ass_01 CheckPartOnTableWait 1<br>Ass_01 Load 1<br>N_6 TravelLoaded 0<br>P_6 UnloadToTote 1<br>G_2 Travel 0 |
| 9 | Come back | 3 Call 1 |

Table 5 shows the cycle work performed by the operator (Work column) and the corresponding high-level scripting language instructions (Steps 7–9 of the methodology). Figure 9 shows a view of the simulation model before and after the start of the simulation experiment.

The final steps of the methodology include the Lean analysis. Thanks to the high-level scripting language used, corresponding to the work language, it is possible to assign value-added attributes to work routines. These are the following attributes:

- VA—Value Added;
- NVA—Non Value Added;
- NVAA—Non Value Added Attackable;
- Other.

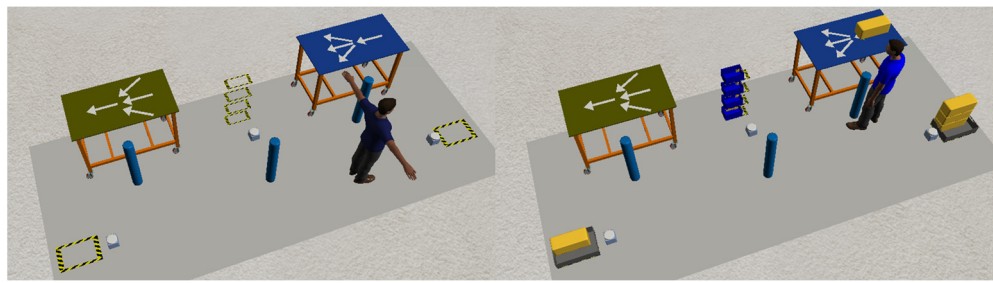

**Figure 9.** View of the simulation model before the start of the simulation (**left side**) and after the start of the simulation experiment (**right side**).

In the simulation approach, we apply the well-established in the business practice, tools for Lean analyses which are actionable for decision-making for remanufacturing, such as:

- Gantt diagrams;
- Analyses of VA;
- Operator load diagrams—so-called Yamazumi Charts, as shown in Figure 10.

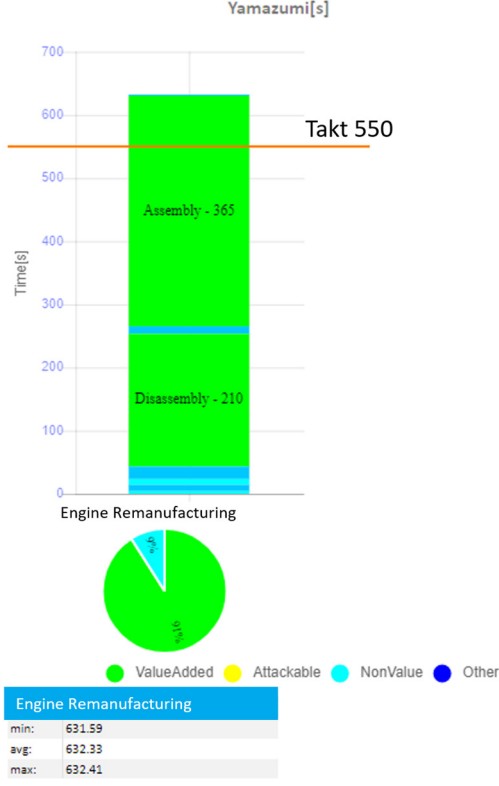

**Figure 10.** Analyses of value-adding operations (VA), and an operator load diagrams—so-called Yamazumi Charts.

### 5.2. Discussion

The presented simulation modeling concept meets the expectations set by the defined scientific and practical challenges. It allows you to shorten the process of creating simulation models. These models are an essential part of digital twins. The reference point for the proposed method are existing simulation modeling methods, based primarily on DES. The essence of DES is to model system operation as a discrete sequence of events over time. Each event occurs over time and represents a change in the system. Applications available on the market are object-oriented programs that provide the user with a set of objects and tools for

creating flow logic (state diagrams, flow diagrams, etc.). They are used to "force" objects to behave like objects in reality. Finally, to match objects, the user must master programming skills. It can be noticed that the development of this software is horizontal, which means that objects are developed and tools for creating logic are developed separately. Building relationships between objects is left to users. This is presented in Section 3.

To sum up, DES offers:

- Event language that is reactive;
- Abstracts, such as state diagrams, flow diagrams, tokens;
- Programming;
- The states of objects to be analyzed.

The users are production engineers, Lean specialists, logisticians, lecturers of manufacturing design courses, factories, logistic processes as well as students of these fields. They use a work language that is proactive. They use Lean, which can be characterized as the language of work—operations that can be given added value attributes. They are very specific and technical—they do not want to use abstracts. They do not want to program, but they want to build digital factories just like real ones. Using DES simulation applications requires them to either increase their competences—that is, learn everything that DES applications require, or employ an intermediary—a simulation modeler who knows the language of events. They, in turn, usually do not know the working language used in real factories. DES/ABS simulation modeling methods require a much longer time to build the model because they only provide basic modeling tools. As mentioned earlier, the construction of relationships and the mapping of real processes are left to the user. The proposed simulation modeling concept significantly shortens this time because it provides the user with modeling tools that are closer to those he uses on a daily basis.

Table 2 in Section 3.1 shows the defined practical challenges. They can only be fulfilled after solving tasks that are defined by scientific challenges. As a summary of this article, on the one hand, it is possible to show practical challenges in relation to the solutions proposed by simulation programs available on the market—Table 6. On the other hand, you can try to present a list of advantages of the proposed approach over DES/ABS.

**Table 6.** Summary of practical challenges in relation to the solutions proposed by simulation programs available on the market.

| Characteristic | Simulation Based on DES/ABS | Proposed Approach—Practical Challenges |
|---|---|---|
| Purpose | Overall | Dedicated to the CEP |
| Types of issues | Systems engineering | Design and analysis of systems implementing the processes: Repair, Renewal, Remanufacturing, Cannibalization, Recycling, Utilization |
| Frequency of use | Single-use | Reusable |
| User | Modeler/Analyst | Production engineer, process engineer, Lean specialist, Logistics specialist |
| Limitations | A high degree of flexibility | Remanufacturing Focus |
| Basic concept | Abstraction | Focused development environment for engineers |
| Terminology | Practical Modeling with Computer Science science; often all new to the engineer and factory staff | Remanufacturing, logistics, Lean; mainly daily deadlines for engineers and factory workers carrying out remanufacturing processes |

**Table 6.** *Cont.*

| Characteristic | Simulation Based on DES/ABS | Proposed Approach—Practical Challenges |
|---|---|---|
| Interface | Data objects, relational tables, scripts, and other data structures | Tables and Actions |
| Time to get results | his in-depth process covers problem definition, model development and testing, data collection, and performance measurement. | Rapid Results (Known System Configuration) |
| Resources required | Many different stakeholders | Minimal Resources: Existing Production Staff with Lean Expertise |
| Knowledge/training required | Requires expertise in simulation, statistics, and software | Low Barrier to Entry: Focus on User Interface and Basic Tools |

## 6. Conclusions

Focusing on a novel simulation modeling approach for remanufacturing processes, this article highlights the synergy between sustainable development and efficient production. Remanufacturing, a cornerstone of sustainability, is integrated with simulation modeling to create a powerful tool. This method empowers organizations to achieve substantial improvements in their remanufacturing processes. It allows for early stage testing of shop floor changes and an investigation of uncertainties' impact. By ensuring effective practices, this approach ultimately yields more sustainable strategies. At the heart of this efficiency lies optimization, encompassing both spatial considerations and process flow.

This approach excels by incorporating lean manufacturing principles to minimize waste across production and remanufacturing. This not only involves designing workflows that promote material reuse but also encompasses:

- Streamlined Processes: Minimizing unnecessary steps and optimizing flow to reduce waste generation.
- Improved Ergonomics: Designing workstations for optimal comfort and efficiency, reducing waste due to fatigue or injury.
- Optimized Material Handling: Reducing unnecessary movement of materials by AGVs, forklifts, or personnel through layout improvements and potentially implementing designated work zones. This not only saves time but also reduces energy consumption associated with transport.
- Process Improvement: Identifying and addressing the causes of product defects, leading to higher first-pass yields and reducing waste materials.

By employing these lean principles, the proposed method fosters sustainability through a multifaceted approach to waste reduction.

**Funding:** This study received support from the SCANDERE project (Scaling up a circular economy business model by new design, leaner remanufacturing, and automated material recycling technologies), which was granted by the ERA-MIN3 program under grant number 101003575. Poznan University of Technology was financially supported by NCBR, National Centre for Research and Development, Poland (No. ERA-MIN3/1/SCANDERE/4/2022).

**Institutional Review Board Statement:** Not applicable.

**Informed Consent Statement:** Not applicable.

**Data Availability Statement:** The raw data supporting the conclusions of this article will be made available by the authors on request.

**Acknowledgments:** This study received support from the SCANDERE project (Scaling up a circular economy business model by new design, leaner remanufacturing, and automated material recycling technologies), which was granted by the ERA-MIN3 program under grant number 101003575. Poznan University of Technology was financially supported by NCBR, National Centre for Research and Development, Poland (No. ERA-MIN3/1/SCANDERE/4/2022).

**Conflicts of Interest:** The author declare no conflicts of interest.

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
