# Peer review of "Scientific and Practical Challenges for the Development of a New Approach to the Simulation of Remanufacturing"

_sustainability, doi:10.3390/su16093857_

Round 1

Reviewer 1 Report

Comments and Suggestions for Authors

1. The manuscript effectively introduces the concept of simulation modeling methods for remanufacturing within a circular economy framework. However, it takes almost half of the content for introduction and literature review with some basic concept definitions such as DT, DES, ABS and Structuring etc. It would be beneficial to provide a brief and concise introduction and literature review for problem statement.

2. The manuscript adequately outlines the key hurdles associated with implementing simulation modeling methods in remanufacturing processes. However, it would be helpful for readers to understand these hurdles by providing some examples in real world (from scientific literature or industry project).

3. It would be valuable to include some example or a specific case study demonstrating the practical application of the proposed simulation modeling approach in real-world remanufacturing scenarios. This would help readers better understand how the approach can be implemented and its potential benefits in improving remanufacturing processes within a circular economy framework.

4. The manuscript looks a combine of lean manufacturing principles and digital twin methods then apply them into remanufacturing. It is hard to tell the innovation in terms of scientific merits and any future potential for industrial applications. The author may need to point out the scientific merits (in terms of knowledge generation) and social impacts (in terms of industrial applications).

Comments on the Quality of English Language

The manuscript's English writing is sufficient for comprehension, albeit with some formatting issues that require attention. For instance, there are two 3.1. scientific challenges and practical challenges

Author Response

Thank you for review

  1. Structure of paper was changed, and section - "2.3 Definition of research gap "was separated to better formulate the problem
  2. Section "5.1 Results - practical example"  was written from scratch along with a practical example.
  3. like point 2
  4. Innovation consists in changing the way of thinking (paradigm), moving from a reactive language (characteristic of DES) to a proactive one, which is the essence of the new approach - this implies the generation of new knowledge in the field of process structuring, a clear separation of the executive level (cycles) from the controlling-managing level. This is the direction of my current and future work.
  5. Manuscript was reformatted and corrected including sections numbering.

Reviewer 2 Report

Comments and Suggestions for Authors

(1) Please describe the novelty of the approach to simulation in details. In addition, please summarize the principles of this method systematically.

(2) This work doesn't seem like a scientific paper. 

Author Response

Thank you for review

  1. Section "5.1 Results - practical example"  was written from scratch along with a practical example
  2. Structure of paper was changed - according to requirements for scientific paper, and section - "2.3 Definition of research gap "was separated to better formulate the problem, and section "5. Results and Discussion" and "6,  Conclusions" have been rewritten.

Reviewer 3 Report

Comments and Suggestions for Authors

In this paper, the concept of a new modeling approach is proposed in order to address the challenges of science and practice. And compare with existing solutions. Overall, the article is organized. In Reviewer's opinion, several issues should be addressed to improve this paper as follows:

1. In the introduction, the author can add a separate paragraph to describe the organizational framework of the paper.

2. Authors need to restructure the article to make it more logical.

3. Authors need to update references.

Comments on the Quality of English Language

The logic of the language is poor.

Author Response

Than you for review

  1. A separate paragraph describing the organizational framework of the work has been added
  2. The article has a changed structure. Section - "2.3 Definition of research gap" was separated to better formulate the problem, Section "5.1 Results - practical example"  was written from scratch along with a practical example.
  3. the literature has been updated
  4. the language has been improved

Reviewer 4 Report

Comments and Suggestions for Authors

The article “Scientific and practical challenges for the development of a new approach to the simulation of remanufacturing” is dedicated to comparative study. The work is well done.

I recommend revision with some improvements.

My recommendations:

1.       The link of ref.1 does not work.

2.       The ref. 32 is not relevant, it is until 2019.

3.       Write the name of the axis in figure 4.

4.       Figure 5 has elements and text that cannot be clearly visualized, and there is also a lot of free space left under the figure.

5.       Figure 6 is the same problem as fig 5.

6.       Figure 7 is the same problem as fig 5.

7.       Before the Conclusions it is recommended to be a section "Results and Discussion", where a more detailed description could be made of the method used and to introduce table 4, which is put to Conclusions.

8.       The conclusions are too general, it is necessary to develop them.

9.       I consider that the introduction and presentation of the concepts applied in the paper are relatively broad in relation to practical challenge.

10.   Please make the table 4 to be in one page.

11.   Reformulate the phrases where the signaled similarity is high.

Author Response

Than you for review

Round 2

Reviewer 2 Report

Comments and Suggestions for Authors

Although the comments have been responded, I still think this work need to be described scientifically.  But at this time, I gave a chance to recommend this work to be accepted.